# The Impact of the Stock Market on Liquidity and Economic Growth: Evidence of Volatile Market

Collin Chikwira * and Jahed Iqbal Mohammed

School of Public Management, Governance and Public Policy, College of Business and Economics, University of Johannesburg, Auckland Park, Johannesburg 2092, South Africa
* Correspondence: collincolchi1913@gmail.com; Tel.: +27-(0)-11-559-7638

**Abstract:** Stock markets serve as a conduit for money and liquidity, which are necessary for economic growth and stability. This study aimed to determine whether stock market impacts are communicated in an economically unstable environment, characterised by volatility, high inflation rates, and political instability. The research used a time series Vector Autoregressive model (VAR) with quarterly data from between 2013 and 2022. The study revealed that there is a positive statistically significant association between the stock market and economic growth at the 10% level. On the other hand, the stock market liquidity has no major influence on Zimbabwe's economic development. As a result, the study advises policymakers to evaluate the rules regulating the stock market carefully and to relax some of the requirements for firms to be listed on the stock exchange. The stock market will become more liquid as a result of this because it will draw more internal and external businesses to being listed. The ZSE should also develop a framework for the gradual implementation of the commodity derivatives exchange as Zimbabwe's substantial mineral reserves and robust agriculture may bring significant income to the country's economy.

**Keywords:** stock exchange; liquidity; economic growth

## 1. Introduction

To sustain economic growth and development, seamless commerce, investments, and capital raising skills of investors, institutions, and governments are required. Hence, exchange markets are critical for economic growth and development (Demirgüç-Kunt and Levine 1996). According to Levine and Zervos (1996), the size of the stock market is an important factor in stabilising the way capital may be obtained in order to support economic development and investments more effectively. Empirically, the link between stock market development and economic growth has attracted considerable attention, as evidenced by the number of studies in the field conducted by academics. They determined that the link is generally positive in developed economies. The question of why such a link occurs remains unresolved: is it technological progress, the industrialisation impact, the liquidity of capital markets, or does the stock exchange drive the economy? As a result, the current study seeks to answer such concerns using Zimbabwean evidence.

According to Demirgüç-Kunt and Levine (1996), the stock market is an important market that connects savers and borrowers, creating liquidity so that corporates, people, and governments may exchange equities as a readily available key component of growth. The financial securities exchanged on the stock exchange are vital for the economy to function properly. For an economy to flourish, funding tools must be widely accessible, which is why the stock exchange is so important. Therefore, the current study will answer the question: what is the impact of the stock market in an unstable economy?

*Background*

The interconnections between exchange markets and economic growth have been intensively argued in academic circles, conferences, and with policymakers in order to

determine which aspects are vital for economic sustainability (Magweva and Mashamba 2016). Overall, researchers have endeavoured to determine whether the presence of an exchange market contributes to the growth of economies and how listed firms benefit from being listed on the exchange. As a result, many researchers have been interested in the influence of the exchange market on the well-being of nations through economic activities that resulted in the rise of their gross domestic product (GDP) (Alajekwu and Achugbu 2012). The researchers' principal attention has been centred on two factors, the first being whether there is a link between stock market activity and economic growth within these concepts. The second feature is the type or, if existent, the direction of the causal connection. For example, the test may determine whether the causation direction of the variables under consideration is unidirectional, bidirectional, or independent. If it demonstrates independent causation, it suggests that the variables, namely stock market development and economic growth, are not linked to or do not cause each other (Şendeniz-Yüncü et al. 2018). Over the years, there has been debate about the connection between organised exchange markets and economic growth. To date, neither side has reached a definite conclusion about the theoretical or empirical link between the two. The supply-leading hypothesis contends that developing financial systems leads to economic growth by channelling savings into investments. According to Hailemariam and Guotai (2014), the functioning of the stock market is a necessary financial instrument for economic growth in any given economy. Furthermore, Bongini et al. (2017) explained that economies with well-functioning financial markets grow faster and ease the financing constraints that affect the expansion of firms and industries that are essential for economic growth. These findings are similar to those of (Elfeituri et al. 2023), and they are evident in the ZSE's mission, which prioritises capital and risk management.

The role that stock markets are thought to play as financial intermediaries, including mobilising savings, reducing risk, and long-term capital investment (El-Masry 2006; Antoniou et al. 2008), has generated controversy regarding the impact of stock market growth over the years. The ZSE is no exception to this rule.

Bhowmik and Wang (2020) concluded that the stock market is a fundamental insulin in the economic activities in the modern world. It is a gauge meter used to test the economy's well-being among a given calamity because it is the first market to send a signal of the growth or cycle of the business trends to the policymakers. Thus, the volatility of stock index returns is an important variable to measure adversity in an economy.

Chaudhary et al. (2020) strengthened the claim that uncertainty in the markets is a view of the volatility of the stock markets, which has the highest bearing on investment and portfolio management analysis. Volatility indicates an economy's instability.

There are 29 stock exchanges in Africa; the Johannesburg Stock Exchange (JSE) is the largest, with a market capitalisation of USD 1.36 trillion, followed by the Nigerian Stock Exchange (NSE), which has a capitalisation of USD 66.7 billion. Furthermore, the Casablanca Morocco Stock Exchange has a capitalisation of USD 65.3 billion (Onyango et al. 2023). Given that the stock exchanges hold substantial amounts of market liquidity, studies linking them to increases in the GDP in the respective countries have found an exponential expansion in their use. According to Levine and Zervos (1996), an effective financial system is a prerequisite for economic progress. The importance of banks and the stock market in an economy was emphasised by Beck and Levine (2004), who also noted that these factors are consistent with the theory. This implies that a sound financial system distributes wealth to productive sectors that require capital to fund the production of goods and services that stimulate economic growth.

The smooth operation of the financial system is particularly important to Zimbabwe's economy. The capital market enables corporations to raise money by issuing shares, while banks are responsible for supporting economic operations by giving loans. Governments issue bonds to ensure economic sustainability. With a market capitalisation of ZWL 741 billion and a market turnover of ZWL199 million as of 2 July 2021, Zimbabwe is one of the oldest stock exchanges in the world. It was created in 1894 (Zimstat 2021).

The stock exchange is where participants may trade shares and other financial securities in a highly regulated and safe atmosphere (Mabilesta 2016). As a result, traders will conduct business with a high degree of confidence and little danger (Chen et al. 2020). The stock market operates as either a primary or secondary market, each governed by specific regulations enforced by the authorities. As a primary market, it is where new shares are issued, enabling businesses to sell their stock to the general public through the process of an IPO (Beck and Levine 2004). Considering their massive stock market valuations and crucial roles in the financial system, policymakers and experts have been compelled to investigate whether they were associated with economic development.

Chen et al. (2020) showed that while long-term productive investment projects need a high level of liquidity commitment, savers do not need to keep their assets for an extended period. As a result, equity markets alleviate this strain by providing savers with an asset that they can rapidly and inexpensively sell (Şendeniz-Yüncü et al. 2018). The current analysis adds empirical information about the stock market and GDP growth to the corporate finance literature. Theoretically, trading in the stock market influences liquidity, risk diversification, business information acquisition, corporate control, and savings mobilisation (Dube 2020).

By marginally altering the quality of the service provided by the currency market, the rate of economic transmission to GDP growth will be influenced positively. There is inconsistency in the empirical data; other studies have discovered that the stock market negatively influences economic growth (Arcand et al. 2015; Demetriades and Rousseau 2016; Rousseau and Wachtel 2011), while others affirm a favourable effect (Demirgüç-Kunt and Levine 1996; Greenwood and Jovanovic 1990; Greenwood and Smith 1997; Levine and Zervos 1998). As a result, the current study intends to retest the influence of stock markets using the dataset of Zimbabwe, a nation characterised by an unstable economy, price volatility, high inflation rates, an uncertain political climate, and an unpredictable economy. The findings of this study are mainly concerned with the sign and question of causation among the variables of the investigation, which include the stock market, liquidity, and economic growth. Furthermore, the study adds to the body of knowledge via corporate finance literature for analysing the influence of exchange trade markets on unstable economies. Additionally, statistical and conceptualisation challenges must be resolved. This is due to the empirical data indicating that the stock market is vital for economic growth. In contrast, previous research used cross-sectional analysis, which suffers from the problem of cross-country growth regressions. This is because the regression analysis assumed that the observations were obtained from the same population but were taken from widely different nations, necessitating a single country analysis. Additionally, the study employed single nation analysis, focusing on Zimbabwe. As countries are dynamic, particularly in this age of the fourth industrialisation, policies change regularly, economies are affected differently, business cycles change regularly, and governments rise and fall. Consequently, the study answers the conceptualisation problem as the coefficients of the regressions must be interpreted cautiously if cross-country analysis is utilised. This is because, when averaged across lengthy periods, these nations are influenced by various developments that may or may not occur concurrently, as indicated above. As a result, aggregation obscures essential elements, occurrences, and disparities between nations. As a result, the study will address the statistical and conceptual challenges by examining Zimbabwe's economic growth using the time series economics model, Vector Autoregressive (VAR), to investigate the link between the stock market and economic growth. In addition, compared to a cross-country study, a single country analysis overcomes the question of causation.

Considering Zimbabwe's economic circumstances, which include high inflation, political instability, unemployment, and economic destabilisation, the current study tries to determine whether stock market liquidity influences economic growth in Zimbabwe as stock markets have been empirically shown to be a major factor in stable nations' economic

growth. In light of this, does the stock market contribute to economic stability regardless of the state of the economy?

## 2. A Review of the Literature: Theoretical Review

### 2.1. Economic Growth Theory

The subject of which factors may determine economic growth and development has frequently been posed within the economic growth debate. It is possible to wonder whether the same factors in the same proportions will still be strong in the future if the process of economic growth is dynamic. Economic development occurs when the economy has more resources accessible to it, when those resources are employed more effectively, or when new resources are brought into the production process (Acemoglu 2012). There are several ways to enhance the quantity of a nation's resources, including expanding the labour force by promoting the immigration of foreign workers and making new land arable. Dinh et al. (2019) argue that the provision of credit to the private sector affects economic growth in the short term and that the money supply has a positive effect in terms of both long- and short-term economic growth.

According to traditional economists, increasing production capacity and investments are the main factors affecting economic growth (Rousseau and Wachtel 2011). Neoclassical economics attributed three factors—land, labour, and capital—as the main drivers of economic expansion in the first half of the 20th century. This was sufficient for capitalist economies, and the more they used them, the greater economic development they experienced. In 1957, the Solow-Swan growth model proposed the idea that technology is a key element in determining economic growth. The efficiency of resources and the quality of outputs are increased by technological advancement, which also increases the understanding of production and its procedures. Different aspects of technological change have an impact on economic expansion. This includes everything from designing, developing, and creating new and different inputs and processes to rearranging the current processes and equipment (Polat et al. 2015). It extends beyond purchases of buildings, machinery, and equipment. Economic growth is the shift over time in a country's total output, or the production of goods and services from one period to the next. It is the increase in the utilisation of resources in societies, whether through the establishment of new factories or the opening of untouched land for use. By annualising the changes in per capita income, growth is thus made respectable (Agarwal 2019).

Sala-i-Martin (2001) added that the following factors are crucial for measuring economic growth: raising start-up finances must be accessible, training of the workforce and education, and well-equipped firms favourable to the economy. This is in response to the growth in academic issues related to adding value to producing goods and services in economies. In addition, the financial system has to be in good working order to allow money to flow to productive sectors, technology, foreign investment, and information accessibility (Acemoglu 2012). There are disagreements between Sala-i-2001 Martin's citation and other theories on the factors mentioned earlier that determine economic growth. The Solow-Sawn theory focused the debate on recent technology advancements. The genesis of economic liberty can be attributed to (Smith 1954; Ricardo 1957; Malthus 1925; Marks 1951). Smith and Ricardo used the law of markets to change the focus of the debate to production. He continued by saying that an increase in output has an impact on the market's growth and size. Ricardo also believes that money serves as a means of exchange for the market's exchange of commodities and services (Levine and Zervos 1998).

According to Schumpeter's theory, economic growth depends on efficient financial systems, a competitive market, and private property that may enable new technologies. A healthy financial system is essential to achieving and maintaining the development of the value of goods and services (Levine and Zervos 1998). According to Levine and Zervos (1998), the financial system's increased liquidity is a byproduct of the exchange market's successful operation. The country's lifeblood flows via the capital market, which is essential for the economy to function efficiently Sill (1997).

### 2.2. Stock Markets Functions

The capital markets' innovative and technical advancements and their adoption of the Fourth Industrial Revolution items have enhanced and increased the efficiency of the stock markets' operations. Artificial intelligence instruments make it simple to access information, trade, engage in resource allocation, and obtain pertinent information for application in the stock market (Dube 2020). To obtain growth from capital investment, it is important to consider whether the new asset will contribute to creating new goods and services rather than just replacing the company's equipment. Therefore, in order for an economy to benefit from the wreckage of stock markets and experience economic progress, it must either directly or indirectly do so by offering services that encourage the development of new commodities (Jecheche 2012). Long-term capital for public and commercial organisations may be among the services offered O'Shea and Davis (2021). The financial system uses stock markets as an intermediary to move resources from surplus units to deficit units.

Napitupulu and Mohamed (2023) explained that the unexpected outbreak of COVID-19 exacerbated the volatility and extreme price shocks in stock markets, implying that the markets are susceptible to activities that affect investors' mobility. It is noted that stock markets are either linear or nonlinear in their characteristics. In such circumstances, the stock markets react to corruption, political issues, violence, and strikes. Thus, if not controlled, it affects the decisions made by investors, thus affecting the liquidity of the stock market and transferring the rubrics to economic growth. Therefore, the current study needs to determine whether the effects of the stock market are communicated in a market characterised by volatility, inconsistent policies, and political instability. The spillover effect of country risk impacts the performance of stock markets.

Samarasinghe (2023) explained that stock markets are intruding into the financial market by becoming a solid option for financing projects and then applying for loan borrowings—further stressing that if the stock market is liquid, it facilities cheap sources of financing needed by corporations. A liquid stock market offers high extended returns by attracting investors, which will affect bank deposits. They concluded that stock market liquidity allows firms to rely more on equity financing than debt from borrowings.

This is an essential component of how the financial system works because it gives businesses a way to raise equity funding rather than debt. The stock market draws domestic and foreign companies to participate, forming large groups and attracting various players who inject more money into the economy, thereby increasing the liquidity. The stock market attracts domestic and international businesses, establishing sizable groupings and bringing in a wide range of participants who will improve the liquidity by injecting more money into the economy. Another advantage is that the stock market combines the little sums saved by hundreds of individuals and grows them into substantial assets.

Chiang and Zheng (2015) explained that in some empirical research, the bid-ask spread and trading cost had been used to measure liquidity. It is recognised that trading volume/turnover fails to capture the trading cost and the price impact per trade, resulting in a high liquidity value during periods of financial crisis. Historically, Chiang and Zheng (2015) witnessed increased trading volumes and drastic declines in market liquidity during crisis periods, such as the 1997 Asian flu crisis, the 1999 Argentine turmoil, and the 2008 credit market crisis. The current knowledge contends that securities with higher trading volume turnover are considered to comprise more liquid stocks. As trading volumes vary in absolute values among different markets, we re-scaled the numbers to make them comparable.

Theoretically, it is asserted that stock markets are highly effective in fostering macroeconomic growth. That is, they encourage liquidity. The stock market allows businesses to obtain desperately needed capital quickly, easily, and affordably, positively impacting capital allocation, investment, and growth (Magweva and Mashamba 2016). They further allow the easy trading of equities, an important element of risk management in investments. By involving international investors, stock markets contribute to the transparency and

strengthening of rules in emerging economies. This is due to the fact that adequate trade ethics, accountability, and unambiguous shareholder rights will be required to preserve investments (Bernal-Ponce et al. 2020).

*2.3. Empirical Literature*

Given that China is the national economy with the most significant rate of development, Pan and Mishra (2018) have examined whether the stock market affects the country's GDP. The study's estimators used the ARDL model and unit root test. The research revealed a destructive long-term relationship between the real sector and the economy. The stock market's unreasonable prosperity and the banking sector's economic bubble were the economic postulates that had been the root cause. According to the present study, their research did not consider China's stock market or short-term economic growth rate.

Further analysis, using the Toda Yamamoto causality test, revealed that economic success had driven the expansion of the Shenzhen B share market. In their conclusion, they attributed the majority of China's economic performance to state-owned monopolies. As a result, the present study will look at a mixed economy.

Qamruzzaman and Wei (2018) analysed Bangladesh's economy by assessing the financial innovation and stock market growth between 1980–2026. They used ARDL and Granger causality tests as their methodology to find the results. The tests of ARDL confirm the long-term relationship among financial innovation, the stock market, and economic well-being. Furthermore, the Granger causality tests yielded bidirectional relationships in both the short- and long-term associations. The findings agree with the theory that financial innovation and financial development are the engines of economic growth.

Shravani and Sharma (2020) also analysed the relationship between selected indicators of the Indian Stock Exchange and the index of industrial production by considering the timeframe of 1996–1997 to 2015–2016. An autoregressive distributed lag estimator and a vector error correction model were used in their analysis to establish the long-term relationship. The study found a long-term relationship between the stock market and economic growth. The policymakers in India are proposing fewer restrictions on the listing requirements so that more companies can take part in the stock exchange.

Nathaniel et al. (2020) explored the Nigerian economy through the RDL approach between 1980–2016 and they found that the stock markets are promoting the growth and development of innovations, which has led to economic growth. The test showed that this was the case in the short term rather than the long term. Only trade openness was insignificant in the short term among the variables tested, and it was revealed that it is significant in the long term; that is, it impacts economic growth if kept on track for an extended period. In addition, it shows that trade openness was negatively associated with economic growth from this study in Nigeria. Furthermore, they recommend policies that would allow more participation in the stock exchange, promoting liquidity, stability, and accountability in the stock exchange market.

Osamwonyi and Osaseri (2020) analysed two economies, Nigeria and South Africa, in the period 1995–2015 using quarterly data. The study employed the Granger causality test and ordinary least squares with panel estimation methods to establish the effect of the stock exchange on economic growth. The results show that the stock exchange does not cause economic growth, and vice versa, in the Nigerian economy.

In Ghana, Borteye and Peprah (2022) used a correlational research design with the use of SPSS to understand the effects of the development of the stock market on economic growth. The study inclined on the market liquidity, size, and capitalisation of the stock market if they transmit the residues to economic growth. Their findings show that stock market liquidity and economic growth have a positive relationship, and the market size has a negative effect on economic growth. Their conclusion stated that stock market development and economic growth had a positive confirmatory relationship in Ghana but were not statistically significant, implying that the study fails to deduce the net effect of the

stock market on economic growth. Their study raised the issue of the increasing number of counters listed on the stock exchange in Ghana.

Islam et al. (2023) investigated the impact of macroeconomic drivers, such as the gross domestic growth rate, inflation rate, and industrial production index, on the Dhaka stock exchange, i.e., DSE 30 index, through the use of statistical techniques such as descriptive statistics, Pearson correlation analysis, and multiple regression analysis. They found a significant and positive relationship between the Bangladesh Stock Market index and the GDP rate. This implies that an increase in the GDP leads to a corresponding increase in the stock market. From their analysis, it is confirmed that the GDP is a key factor that affects the performance of the stock markets in Bangladesh. Thus, an increase in the GDP means increased investment opportunities, thus driving the stock markets upwards. Furthermore, the results also revealed that the stock market positively influences the economy by creating new job opportunities and encouraging entrepreneurship. In addition, Elfeituri et al. (2023) found that there is a positive relationship between the stock market and economic growth in the Gulf countries, which was influenced by the stock turn over and stock capitalisation.

Magweva and Mashamba (2016) used the Zimbabwean context to analyse the impact of the stock exchange through the data covering the period between 1989 and 2014. They employed the Vector Error Correction Model (VECM) approach. From their analyses, they observed that there is a negative long-term relationship between the variables. The authors believe that the authorities must entice small and medium enterprises to list and participate on the stock exchange to promote more participation, bringing more liquidity to the market. According to Magweva and Mashamba (2016), they believe that firms are raising finances from other sources rather than the stock exchange; therefore, they recommended that corporations focus on other avenues for economic growth. Thus, the current study added a new variable into the model that captures the economic activities measured by international trade and the component of the liquidity of the stock exchange.

Mawanza et al. (2020) analysed the Zimbabwean stock exchange impact with a dataset from 1980 to 2018. In the analysis, they employ the ordinary least squares technique for data analysis for inference. Their results concluded that the stock exchange and economic growth had blended effects. Further research confirms a positive relationship between stock market capitalisation and foreign direct investment, which benefited the economic development.

Shoko et al. (2020) also presented a different view of Zimbabwe's economy by analysing the stock market using the ARDL error correction model. Their study incorporates money supply, interest rates, exchange rates, and GDP as their variables. In the long term, only the exchange rate had a significant positive relationship with the rate at which the economy grew. In addition, the occurrence of inflation reflects that it did not confirm a statistical relationship with economic growth. While GDP, money supply, and interest rates show an insignificant relationship with the economy's growth, the causality relationship shows a bidirectional between the stock market and real GDP and a unidirectional between inflation and the stock exchange, which ran from the stock market to the interest rate. With a close look at the results of Shoko et al. (2020), the authorities should consider policies that promote, and are essential to, stabilising the fundamental macroeconomic environment in Zimbabwe.

The development of strategies that are critical for innovations that lead to the launch of new goods and services are the core policy implications. Thus, the rate at which economies grow due to debris extracted from the developments on the stock exchanges has yielded mixed results. In addition, the literature shows that the causality ran from both sides in different periods and in nations from developing or developed countries. Therefore, this study provides an overview of whether the relationship exists in a Zimbabwean context, which will in turn provoke policy recommendations that might assist economic growth through stock market developments. This led to the postulation of the hypotheses, as explained below:

**H0₁.** *There is no positive relationship between the stock market, liquidity, and economic growth in unstable economies.*

**H1₁.** *There is a positive relationship between the stock market, liquidity, and economic growth in unstable economies.*

### 3. Methodology

The study is purely quantitative; therefore, the analyses employed a vector autoregressive (VAR) model to understand the direction of causality among the variables.

The data were sourced in quarterly frequencies for the period between 2013 and 2022. The study used five variables, which are economic growth (GDP per capita), stock market size (market capitalisation), market efficiency (turnover ratio), stock market liquidity (stock market turnover), and economic activities (trade as a percentage of GDP).

As a rule of thumb, in VAR analysis, the equation should contain k lag values of the variable in the equation, for instance, real domestic growth (GDP) and stock market size (STK) as the main variables are illustrated in the model below. In this case, the equations are estimated by OLS, as follows, in a simple regression of the equation:

$$\text{GDP}_t = \alpha + \sum_{i=1}^{k} \beta_i \, \text{GDP}_{t-i} + \sum_{i=1}^{k} \gamma_i \, \text{STK}_{t-i} + \mu_{1t} \tag{1}$$

$$\text{STK}_t = \sigma + \sum_{i=1}^{k} \beta_i \, \text{GDP}_{t-i} + \sum_{i=1}^{k} \gamma_i \, \text{STK}_{t-i} + \mu_{2t} \tag{2}$$

where $\mu_{1t}$, $\mu_{2t}$ are the stochastic error terms, called impulse or innovations, or shocks in the VAR system, k lag length, $\alpha$, $\sigma$ intercept $\gamma_i$, and $\beta_i$ are the short-term coefficients.

VAR is the most appropriate technique for this study because the research wishes to establish the causality and direction of causation among the studied variables. In addition, it helps to relate the dynamic behaviour of economic and financial time series and forecasting (Gujarati 2009). Forecasts using the VAR model are fairly flexible because they can be made conditional on the potential future paths of specified variables in the model (Vo et al. (2019).

The study first performs a stationarity test using the Augmented Dicker Fuller Test (ADFT) because the dataset is a time series. To avoid spurious results in the VAR model, the first step is to test the stationarity of the series. If the data are not stationary, the regression results cannot be inferred (Heino 2005), implying that the series data generating process does not involve zero.

In econometrics, stationarity is very important when using time series data; thus, the dataset must be stationary. For time series data to be used for inference, the first step is to perform stationarity tests to establish the dataset's status. It is not easy to generalise the series to other periods (Baumohl and Lyocsa 2009). If the data is non-stationary, it cannot be used for forecasting purposes, and it will have little practical use. In addition, if the dataset has two or more non-stationary time series, the regression analysis results generated will be spurious or nonsense regression (Gujarati and Porter 1999). The results will be unreliable and cannot be used in policy analysis and structural inference. For the reasons stated, testing for stationarity in the time series is important.

### 4. Data Sources and Issues

Data for 2013 to 2022, extracted in quarterly series, were used in the study. In the model, there are five variables, which are economic growth rate (GDP per capita), stock market size (STK), stock market efficiency (STF), stock market liquidity (STQ), and economic activities (EXP). The data sources are the World Bank data indicators and Reserve Bank of Zimbabwe, complemented by the ZIMSTA, which was the custodian of all the data in Zimbabwe.

**Economic growth** is the dependent variable. The real GDP per capita growth rate was used as a proxy for economic growth in the study, following (Bernal-Ponce et al. 2020); Magweva and Mashamba (2016); (Şendeniz-Yüncü et al. 2018). The real GDP measures the domestic production per person; thus, this study assumes it will reflect the productivity and welfare of the economy.

**Stock Market Size (SMK)** The size of the stock market impacts economic activities. The size means more players, thus exhibiting many firms in an economy, implying a greater production of goods and services. It is proxied by stock market capitalisation as a ratio to the GDP with references from (Alajekwu and Achugbu 2012; Levine and Zervos 1996; Magweva and Mashamba 2016). Market capitalisation is the value of all the listed stocks on an exchange market. It is measured as a fraction of the market capitalisation to the GDP; the expected sign for the results is positive and should be statistically significant. The study used this variable to capture the growth of the stock exchange and its ability to mobilise capital and diversity (Levine and Zervos 1996).

**Stock market liquidity (STL)**; this variable captures the stock market's liquidity impact. It is calculated as the ratio of the value of total shares traded to market capitalisation as used by (Beck and Levine 2004; Kanetsi 2014; Magweva and Mashamba 2016). Stock market liquidity complements the market size because the size and liquidity must be related.

**Stocks turnover ratio (STF)** (Demetriades and Rousseau (2016); (Magweva and Mashamba 2016) and Beck and Levine (2004)) is the measure of the value of shares traded on the exchange as a ratio of the nominal GDP, which will capture the level of economic activities in an economy.

**Economic activities (EXP)** are a country's total exports and imports over time (Odhiambo 2014). It is a fundamental variable in the model; it has been included to capture the role of trade in economic activities. The expected sign is positive and statistically significant. The economic activities are calculated as the ratio of the total exports and imports to the GDP. It has two limitations: it does not adjust for the size of the economy; it does not directly incorporate the effects of tariffs or protection on economic growth.

*Model Specification*

The dependent variable in the VAR system is a function of its lagged values and the lagged values of the other variables in the model (Gujarati 2009). Furthermore, the model is specified in levels because specifying it in differences leads to model misspecification (Anderson and Hsiao 1982). The VAR system comprises a set of linear dynamic equations, in which each variable is stated as a function of an equal number of lags (k) of itself and all other variables in the system (Gujarati and Porter 1999). Possessing five variables necessitates estimating five VAR models, and all of the variables are endogenous, which means they are determined inside the system.

To test the statistical relationships among variables in econometrics, the first stage is to establish the model, followed by testing the stationarity among the variables and testing the cointegration or determining the long-term relationship, and finally, conducting the VAR model (or VECM if the variables are cointegrated)

The general model specification is represented as follows, which follows the neoclassical growth model:

$$\text{InGDP}_t = \varphi + \sum_{i=1}^{k} \beta_I \, \text{InGDP}_{t-1} + \sum_{j=1}^{k} \lambda_r \text{InSMK}_{t-j} + \sum_{m=1}^{k} \vartheta_m \text{InSTL}_{t-m} + \sum_{r=1}^{k} \varnothing_j \text{InSFF}_{t-r} + \sum_{r=1}^{k} \varphi_f \text{InEXP}_{t-r} + \mu_{1t} \quad (3)$$

$$\text{InSMK}_t = \alpha + \sum_{i=1}^{k} I \text{InGDP}_{t-1} + \sum_{j=1}^{k} \lambda_r \text{InSMK}_{t-j} + \sum_{m=1}^{k} \vartheta_m \text{InSTL}_{t-m} + \sum_{r=1}^{k} \varnothing_j \text{InSTF}_{t-r} + \sum_{r=1}^{k} \varphi_f \text{InEXP}_{t-r} + \mu_{2t} \quad (4)$$

$$\text{InSTL}_t = b + \sum_{i=1}^{k} I_i \, \text{InGDP}_{t-1} + \sum_{j=1}^{k} \varnothing_j \text{InSMK}_{t-j} + \sum_{m=1}^{k} \vartheta_m \text{InSTL}_{t-m} + \sum_{r=1}^{k} \varnothing_j \text{InSTF}_{t-r} + \sum_{r=1}^{k} \varphi_f \text{InEXP}_{t-r} + \mu_{3t} \quad (5)$$

$$\text{InSTF}_t = \lambda + \sum_{i=1}^{k} I \text{InGDP}_{t-1} + \sum_{j=1}^{k} \lambda_r \text{InSMK}_{t-j} + \sum_{m=1}^{k} \vartheta_m \text{InSTL}_{t-m} + \sum_{m=1}^{k} \varnothing_j \text{InSTF}_{t-r} + \sum_{r=1}^{k} \varphi_f \text{InEXP}_{t-r} + \mu_{4t} \quad (6)$$

$$\text{InEXP}_t = \Omega + I \text{InGDP}_{t-1} + \sum_{j=1}^{k} \lambda_r \text{InSMK}_{t-j} + \sum_{m=1}^{k} \vartheta_m \text{InSTL}_{t-m} + \sum_{m=1}^{k} \varnothing_j \text{InSTF}_{t-r} + \sum_{r=1}^{k} \varphi_f \text{InEXP}_{t-r} + \mu_{5t} \quad (7)$$

where InGDP is the economic growth of a country measured by the GDP, lnSMK is the measure of the size of the stock exchange, InSTL is the measure of the stock market liquidity, InSTF is the measure of the stock market ratio (efficiency), InEXP is the measure of the economic activities with the rest of the world, k is the lag length, and $\beta_i$, $\varnothing_j$, $\vartheta_m$, $\lambda_r$, $\varphi_f$ are the short-term dynamic coefficients of the model, $\mu_{1,2,3,4,5}$, residuals (stochastic error terms, often called impulses or innovations or shocks).

## 5. Results

The gross domestic product per capita (GDPk) is a dependent variable in Table 1 that measures a nation's growth and development. It has a mean value of 4.11 and a standard deviation of 0.12. The average size of the stock market (SMK) was 7.06, with a high dataset variation from the mean of 1.56 standard deviations. The stock market ratio was 13.04 with a standard deviation of 0.63, while the exports and imports ratio was 1.22 with a standard deviation of 0.83. It was the relationship between the variables in the dataset. The gross domestic product per capita is shown in Table 1 as a dependent variable that measures growth.

**Table 1.** Data Description.

| Variable | Obs | Mean | Std.Dev | Min | Max |
|----------|-----|------|---------|-----|-----|
| GDPk | 44 | 4.11 | 0.12 | 3.91 | 4.33 |
| SMK | 44 | 7.06 | 1.56 | 3.64 | 10.01 |
| STL | 44 | 8.04 | 0.86 | 0.21 | 4.16 |
| STF | 44 | 13.04 | 0.63 | 11.91 | 14 |
| EXPt | 44 | 1.22 | 0.83 | −2.3 | 2.39 |

### 5.1. Stationarity Results

The time series data were tested for stationarity as follows; the null hypothesis and alternative hypothesis were:

**H0$_2$.** *The variables are not stationary or have a unit root.*

**H1$_2$.** *The variables are stationary.*

To avoid erroneous results, stationarity was confirmed as a precondition for dealing with the time-series data for analysis. The Augmented Dickey-Fuller test (ADF) for the unit root examined the relationship between Zimbabwe's stock market and economic growth. Baumohl and Lyocsa (2009) claimed that the ADF test is superior to the generic Dickey-Fuller test as it tests higher-order autoregression. The level I data were non-stationary (0). Consequently, the variables were converted to the first difference I (1) for it to be stationary. Table 2 displays the ADF test statistics for the model variables at level I (0) and the first difference I (1).

As the test statistics for all the variables were less than the 5% critical value, the results for both levels were not rejected. As a result, the variables are not stationary. Furthermore, when the first difference was tested, the results showed that they were more significant than the 5% crucial limit. At the 5% significance level, the non-stationarity null hypothesis of the first difference is rejected, and the alternative hypothesis is accepted. As a result, the first difference results from the series are stationary. The ADF test decision criterion rejects the alternate hypothesis if the test statistics are less than the 5% critical value, and accepts the alternate hypothesis if the test statistics are more significant than the 5% critical value.

According to the results in Table 2, all of the test statistics at the first difference are greater than the 5% critical threshold, indicating that the null hypothesis is rejected and the variables are stationary.

**Table 2.** Stationarity Test Results.

|  | ADF TEST Z(t) | 5% CRITICAL VALUE |
|---|---|---|
| H0$_2$: The level of the variable is non-stationary |  |  |
| GDPK | 1.618 | 3.000 |
| SMK | 1.892 | 3.000 |
| STF | 0.235 | 3.000 |
| STL | 1.762 | 3.000 |
| EXPt | 0.435 | 3.000 |
| H1$_2$: The first difference of the variable is non-stationary |  |  |
| In(GDPK) | 5.517 | 3.600 |
| In(SMK) | 4.314 | 3.600 |
| In(STF) | 5.635 | 3.600 |
| In(STL) | 4.123 | 3.600 |
| In(EXPt) | 3.750 | 3.000 |

Source: Raw data from STATA.

## 5.2. Empirical Results

The study used the VAR model as an estimator because the objective was to establish the causal effect of the stock market on economic growth and simulate the shocks to the system, and at the same time, to trace the effects of the shocks on the endogenous variables.

Table 3, below, summarises the model equations' estimated results from the STATA. The results extracted from the STATA show that the coefficient is positive at 10%, and is statistically significant between the stock market and economic growth. The positive relationship was also observed by Jecheche (2012). The result implies that the stock market had an impact on the growth of the economy. The relationship is in agreement with the theory postulated by (Beck and Levine 2004; Levine and Zervos 1996), which hypothesises that the stock market drives the economy; that is, it is the source of capital expenditure funds. Furthermore, for corporations to finance their investments and growth opportunities, an efficient source of capital is paramount; thus, companies can raise the required amounts easily.

**Table 3.** VAR Estimates—Short-term effects of the stock market on Economic Growth.

| VARIABLES | GDPGK | SMK | STF | STL | EXPt |
|---|---|---|---|---|---|
| L.GDPG | 0.726 *** | −215.4 | 1.470 *** | −6.262 | 1.088 *** |
|  | (0.267) | (230.1) | (0.34) | (4.468) | (0.183) |
| L2. GDPK | −0.244 | 189.4 | −0.676 ** | 5.91 | 0.921 *** |
|  | (0.218) | (188.3) | (0.278) | (4.419) | (0.15) |
| L.SMK | 0.000568 * | 0.538 ** | −0.00035 | 0.714 *** | 0.001000 *** |
|  | (0.00031) | (0.268) | (0.0004) | (0.132) | (0.00021) |
| L2.SMK | 0.000522 | 0.23 | −0.00025 | −0.185 | 0.00112 *** |
|  | (0.00056) | (0.483) | (0.00071) | (0.126) | (0.00038) |
| L.STF | 0.365 *** | 46.86 | −0.592 *** | −4.989 ** | −0.0157 |
|  | (0.113) | (97.98) | (0.145) | (2.515) | (0.0779) |
| L2.STF | −0.211 * | 139 | −0.722 *** | 6.053 ** | −0.156 ** |
|  | (0.111) | (96.24) | (0.142) | (2.519) | (0.0765) |
| L.LEXPt | −0.700 * | −88.36 | 0.305 | 0.1 | 0.418 * |
|  | (0.358) | (308.8) | (0.456) | (0.0664) | (0.245) |
| L2. EXPt | 0.692 | 11.92 | 2.692 *** | −0.145 * | 0.730 ** |
|  | (0.472) | (407.9) | (0.602) | (0.0766) | (0.324) |
| L.SKL | 0.235 | 6.86 | −0.592 *** | −4.985 * | −0.0167 |
|  | (0.213) | (7.98) | (0.145) | (2.515) | (0.0779) |
| L2.SKL | −0.145 | 139 | −0.773 | 6.053 ** | −0.153 * |
|  | (0.132) | (96.24) | (0.132) | (2.519) | (0.0745) |
| Constant | 3.791 | −9.541 | 106.3 *** | −8.786 *** | 32.65 *** |
|  | −6.808 | (5.877) | (8.678) | (2.854) | (4.67) |

Standard errors in parentheses, *** $p < 0.01$, ** $p < 0.05$, * $p < 0.1$.

The results show that there is a unidirectional effect on the stock market and economic growth through the use of Zimbabwean data. This was seen in the positive relationship between the variables running from the stock market to the development of the economy. Zivengwa et al. (2011) also observed the same effect with the Zimbabwean dataset.

The economic growth and stock market ratio had a positive statistical relation, which is significant at the 1% level. This implies that economic growth influenced the smooth running of the stock market. This was observed in both lags of economic growth, which are statistically significant at 1% and 5%, respectively.

International trade is statistically significant at a 1% to stock market ratio, which implies that economic activities promote economic growth; thus, the movement of goods and services within and outside of the nation adds value to the GDP. That is, enterprises will be able to produce and raise capital requirements inside their capital market and encourage exports and imports. This is consistent with the recently implemented Zimbabwe Auction system, which allows the availability of much-needed foreign currency in the nation for enterprises to conduct business.

The size of the stock market affects the liquidity, and it is positive at a 1% significance level. This implies that the stock market's rise has an impact on the market's liquidity. Furthermore, the economic growth variable had a favourable impact on significant economic activity at the 1% level. The stock market size is also statistically important, contributing 1% to an economy's foreign operations growth. However, the stock efficiency had a detrimental influence on the country's export and import operations, which is significant at the 5% level.

*5.3. Granger Causality Test*

The Granger causality test was used to determine the direction of causality among the variables. The Granger causality test was estimated to further understand the direction of the causal association between stock market growth and economic growth. First, there is unidirectional causality between stock market growth and economic growth in the short term, implying that the stock market causes economic growth but has no effect in the opposite direction. There was a bidirectional correlation between stock market efficiency and economic growth. The findings also demonstrate bidirectional causality between stock market growth and market efficiency. As seen in Table 4, there is also a bidirectional causation relationship between stock market growth and stock market liquidity.

**Table 4.** Granger causality test.

| NULL Hypothesis | chi$^2$ |
|---|---|
| SMK does not granger cause GDPK | 11.172 *** |
| STF does not granger cause GDPK | 12.871 *** |
| STL does not granger cause GDPK | 13.867 *** |
| GDPK does not granger cause SMK | 4.756 |
| STF does not granger cause SMK | 4.603 |
| STL does not granger cause SMK | 5.6216 |
| GDPK does not granger cause STF | 79.492 *** |
| SMK do not granger cause STF | 50.832 *** |
| STL does not granger cause STF | 135.84 *** |
| GDPK does not cause STL | 78.317 *** |
| SMK does not granger cause STL | 23.717 *** |
| STF does not granger cause STL | 54.025 *** |

Standard errors in parentheses *** $p < 0.01$, ** $p < 0.05$, * $p < 0.1$.

*5.4. Post-Estimation Diagnostic Tests*

After the estimation and testing of the time series model with VAR, the results were diagnostically tested for the model's autocorrelation, stability, and reliability, and are presented below.

### 5.4.1. Autocorrelation Test

The null hypothesis (H0$_3$) demonstrates that the lag order has no autocorrelation. The autocorrelation has been tested using two lags. Table 5, the results of the autocorrelation test. The p-value at lag one is 0.616, which is higher than the 5% critical value. As a result, the null hypothesis (H0$_3$), which states that there is no autocorrelation, is accepted. The p-value for Lag 2 is 0.342, which is higher than 5%. As there is no autocorrelation, the null hypothesis (H0$_3$) is again accepted, meaning that the model is desirable and should be used.

**Table 5.** Autocorrelation testing: Lagrange-multiplier test.

| Lag | chi$^2$ | Df | Prob > chi$^2$ |
|---|---|---|---|
| 1 | 13.762 | 16 | 0.616 |
| 2 | 17.000 | 16 | 0.342 |

H0: no autocorrelation at lag order.

### 5.4.2. Residual Normality Test

To determine whether the residuals are regularly distributed, the Jarque-Bera test was applied (Jarque and Bera 1987). Table 6 shows the results of the residual normality test. The GDPK target variable has a p-value of 0.44, which is higher than the 5% threshold. The null hypothesis (H0$_3$) is therefore accepted. The residual is regularly distributed, making the model suitable. Every variable has a normal distribution. Thus, the model is recognized and used for the stock market growth analysis of economic growth.

**Table 6.** Normality testing: Jarque-Bera test of normality.

| Equations | chi$^2$ | Df | Prob > chi$^2$ |
|---|---|---|---|
| D.GDPk | 8.700 | 2 | 0.444 |
| D_SMK | 1.355 | 2 | 0.508 |
| D.STF | 0.462 | 2 | 0.793 |
| D.STL | 1.749 | 2 | 0.417 |
| D.EXPt | 1.862 | 2 | 0.235 |
| ALL | 12.265 | 10 | 0.140 |

### 5.4.3. Stability Tests

All of the computed VAR models are stable as all of the unit roots are contained within the unit circle in Figure 1 shows the results of stability test; thus, VAR meets the stability criteria. As a result, the results of the impulse response function tests can be utilised to make inferences about the economic growth dynamics.

### 5.4.4. Impulse Response Test

Following the estimation of the VAR model, the impulse response function was retrieved and analysed, as shown below. The impulse response function is a helpful tool in econometrics as it tracks the effects of shocks on the variables when there is a percentage change in another variable.

The findings of the impulse response function, which were taken from STATA 15, are shown in Figure 2. Take, for example, the Var-basic, GDPK/SMK, which denotes that GDPK is the impulse and SMK is the reaction, to make the interpretation of the results as simple as possible. As a result, it indicates how the system's SMK variable reacts to a 1% standard shock to the GDPK variable. In the example, SMK is the response variable, while GDPK is an impact variable. The impulse response function was analysed following STATA 15 extracts to track the shocks in the variables. The impulse variable is listed first, followed by the response variable.

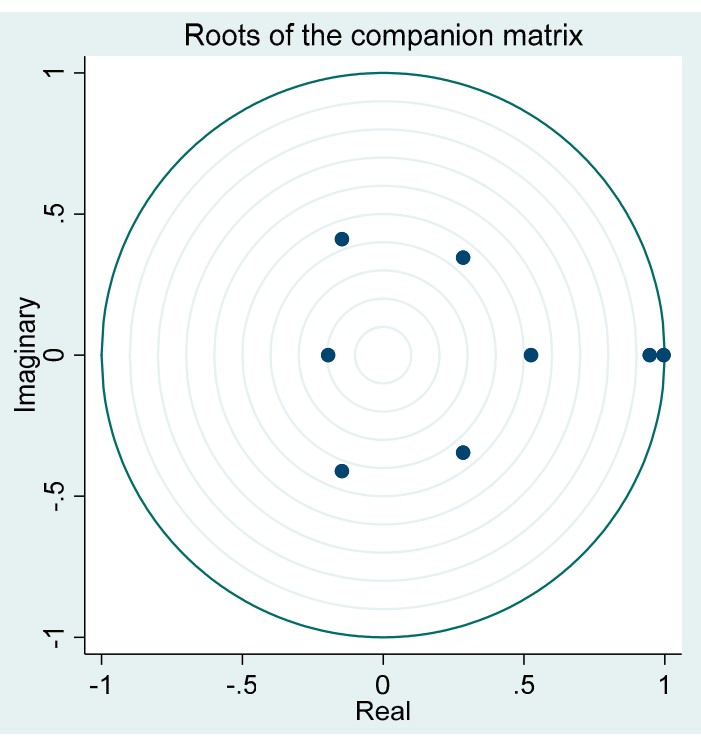

**Figure 1.** Varstable Graph.

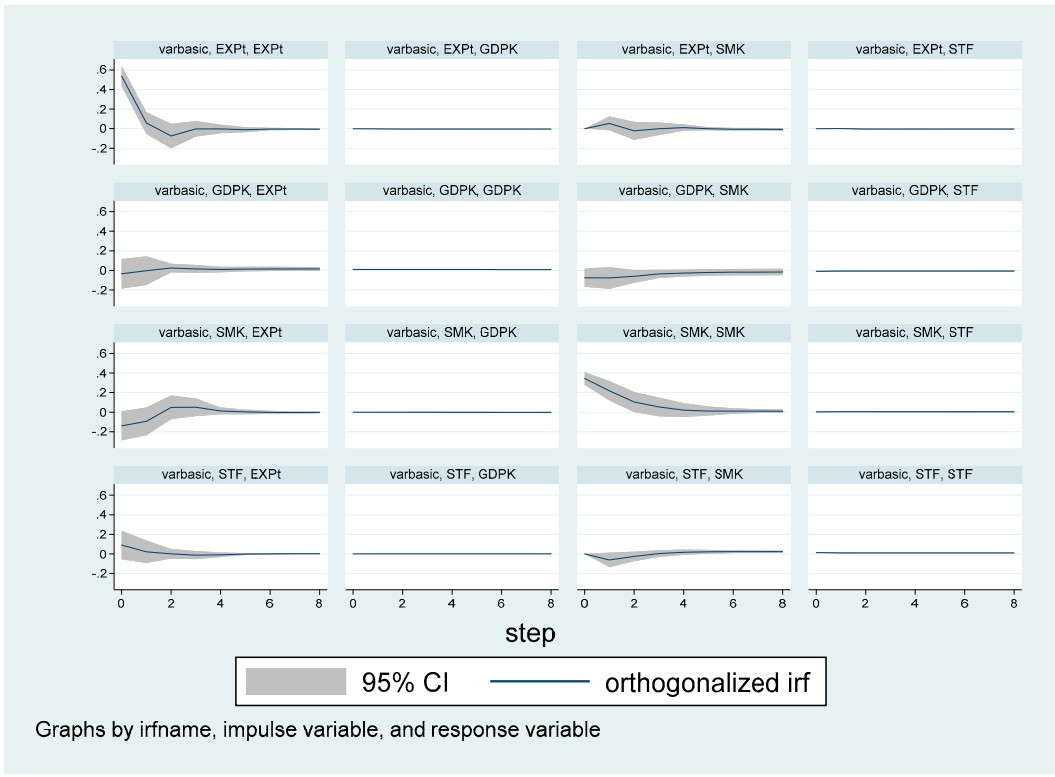

**Figure 2.** Impulse response function tests.

**Varbasic, GDPK, GDPK:** from the extract in Figure 2, the response of the economic growth (GDPK) to its shocks demonstrates that it was constant, i.e., it did not respond favourably or unfavourably to the shocks.

In the long term, the stock market's response to its shocks is favourable, according to the extract from **Varbasic, SMK, and SMK.** The stock market responded favourably to

short-term shocks in the stock market between period 1 and period 4 and between period 4 and period 8; the size of the stock market remained steady and did not respond either favourably or unfavourably to these shocks.

The stock market efficiency variable remained steady over time despite the shocks it applied to itself (**Varbasic, STF, STF** from the results).

**Varbasic, EXPt, EXPt:** the export and import variables, which proxied the economic activity, responded favourably to their shocks from periods 0 to 2. Exports and imports declined from period 2 to period 4 before remaining stable from period 5 to period 8, when they responded neither positively nor negatively to its shock.

**Varbasic, GDPK, and SMK**, the extract of the stock market's reaction to economic growth, show that if the shock of economic growth is a 1% standard deviation, the stock market will decline in periods 1 to 2 and remain constant in periods 2 to 8; that is, it will not be impacted by the growth of the real economy in the long run. If the economy grows by 1%, the stock market will not respond positively or negatively—it will remain the same. This means that the stock market is independent of economic growth; if the economy expands, the size of the stock market will not change.

**Varbasic, SMK, and GDPK** are the extracts of the impulse reactions to stock market shocks and responses to economic expansion. According to the findings, the short-term growth of the stock market is constant, with a 1% standard shock to the (GDPK) economy. The size of the economy did not change in response to a slide shock in the stock market's growth, which eventually led to a decrease in the size of the stock market and a stock market shock. This is due to the absence of any favourable or unfavourable reactions.

Figure 2 shows the stock markets' dynamic response to other variables. All other variables in the system have a positive reaction to shocks. That is a 1% change in the economy, the efficiency of the stock market, and the positive effects of exports and imports on the stock market, which suggest that increasing the stock market's size impacts its efficiency and output, both of which enhance trade. While the shock adversely impacted the exports and imports to the size of the stock market, the dynamic response of the economy's growth was also continuously affected by shocks to the effectiveness of the stock market.

## 6. Discussion

The study found that there is a positive link, although weak, between the stock market and economic growth. Similar findings of positive relationships were also observed in the survey by (Jecheche 2012). Elfeituri et al. (2023) concluded that it is the stock market turnover ratio and market capitalisation that had a positive influence on economic growth, while in this paper, it is the size of the stock market that was most significant. Borteye and Peprah (2022) concluded that, in Ghana, the stock market had a positive effect on the development of the economy because firms use the market to secure funding for growth, which leads to economic activities. The weak significant relationship between the stock market and economic growth translates to the sensitivity of the stock market to market fundamentals. This is evidenced in the study by (Napitupulu and Mohamed 2023), which implies that stock markets are either linear or nonlinear in their characteristics. Under such circumstances, the stock markets react to corruption, political issues, violence, and strikes. Furthermore, if not controlled, it affects the decisions made by investors, thus affecting the stock market's liquidity and transferring the rubrics to economic growth. Therefore, the study exhibits a positive relationship between the stock market and economic growth.

The size of the stock market affects the liquidity, and it is positive at a 1% significance level. Therefore, the null hypothesis is rejected, and the result instead supports the alternative hypothesis of a positive relationship between the stock market and economic growth. This implies that the stock market's rise has an impact on the market's liquidity. Samarasinghe (2023) supported the idea that stock markets are intruding into the financial market by becoming a solid option for financing projects and then applying for loan borrowings—further stressing that if the stock market is liquid, it facilities cheap sources

of financing needed by corporations. A liquid stock market offers high extended returns by attracting investors, which will affect bank deposits. The study concluded that stock market liquidity allows firms to rely more on equity financing than debt from borrowings. Samarasinghe and Uylangco (2021) also highlighted that capital lending has reduced bank loans because it is cheaper for firms to fund their projects through a highly liquid stock market. It is postulated that investors are willing to lend through the stock exchange if the stock market is liquid; thus, the cost of raising equity capital is relatively lower than borrowing through the banks.

## 7. Concluding Remarks

Stock markets have sparked contentious discussions in the financial world, and regulators and policymakers have mixed empirical evidence about the links between the variables. The impact of stock market expansion on economic growth is not universally agreed upon. Consequently, the goal of the current study was to shed some light on the impact of the stock market on economic growth. The study used quarterly datasets from the first quarter of 2013 through to the fourth quarter of 2022 to examine the correlation between stock market liquidity and economic growth indicators. The findings show little correlation between stock market size and economic growth. According to the estimated model, a one-way causal relationship between the stock market and economic growth in the short term is positive and statistically significant to 10%. The stock market liquidity variable has a 1% statistical significance, indicating that it impacts economic expansion.

According to the study's findings, to effectively promote the stock market in Zimbabwe, policymakers must create and put into place measures that would remove obstacles to market capitalisation and liquidity. The report advises policymakers to evaluate the stock market rules in-depth and to relax some of the criteria for businesses to be listed on the stock exchange. More internal and external companies will want to list, which will subsequently increase the liquidity of the stock market.

The impact of the stock market in the business fraternity is crucial for managers to carefully monitor stock market trends and movements to identify potential risks and opportunities because the stock market reacts to the information in the market they operate within. For firms to withstand market shocks, the management should ensure adequate financial resources by promoting savings, diversifying with liquid assets, and providing access to emergency funding facilities in order to maintain operations when cash flows are irregular. Thus, managers in a volatile economy should monitor market trends, manage financial resources effectively, hedge their portfolios against market risks, and adapt their business strategies to mitigate the impact of market volatility on their business operations and achieve sustained growth.

For recommendation, the ZSE should develop a framework for the gradual implementation of the commodity derivatives market, pointing out that an economy can benefit significantly from Zimbabwe's substantial mineral reserves and universal-level agribusiness. Moreover, stock market discoveries and developments can raise productivity, which impacts economic growth. For instance, if the ZSE adopts the derivatives market, farmers could use the commodity derivative instruments to hedge and start farming, knowing their exact prices because they have contracts with their product's consumers before they begin. As a result, the increased productivity and producer price security will contribute to the anticipated increase in economic growth. Topcu and Gulal (2020) stated that extensive policy measures are critical for rejuvenating the stock markets. They concluded that the outcomes of tight policies implemented by a number of monetary authorities yielded positive results, distorting the stock markets in emerging markets due to COVID-19.

Although the assessment was for a single country, limitations cannot be avoided. Firstly, the study did not factor in the effects of COVID-19, which were characterised by lockdowns and the closure of businesses, which impacted the stock market and economic growth. Secondly, the political outbreaks of violence that occurred during some of the selected years of study affected the operations of the economy and the stability of the

financial markets; this issue was not considered. Therefore, it is expected that further research should capture the variable that accommodates political violence and corruption to solve the limitations of the paper.

**Author Contributions:** Conceptualization, C.C. and J.I.M.; methodology, C.C.; software, C.C.; validation, C.C. and J.I.M.; formal analysis J.I.M.; investigation, C.C. and J.I.M.; resources, J.I.M.; data curation, C.C.; writing—original draft preparation, C.C.; writing—review and editing, C.C. and J.I.M.; visualization, C.C.; supervision, J.I.M.; project administration, C.C. All authors have read and agreed to the published version of the manuscript.

**Funding:** This research received no external funding and the APC was funded by the University of Johannesburg.

**Institutional Review Board Statement:** Not applicable.

**Informed Consent Statement:** Not applicable.

**Data Availability Statement:** The data presented in this study are available on request from the corresponding author.

**Conflicts of Interest:** The authors declare no conflict of interest.

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
