# Peer review of "The Impact of the Stock Market on Liquidity and Economic Growth: Evidence of Volatile Market"

_economies, doi:10.3390/economies11060155_

Round 1

Reviewer 1 Report

(see attached)

Author Response

Response to Reviewer 1 Comments

  1. The title of the article is not clear. it is hard to find out the precise causality among the related variables

Response Point 1:

  • The title changed to: The Impact of Stock Market on Liquidity and Economic Growth: Evidence of Volatile Market.
  • We have reworded the title.

  1. No article after 2020 has been cited in this study.

Response Point 2:

  • Updated with new citations after 2020
  • Included 2021, 2022, and 2023 citations
  1. Having defined equations 4-7, equations 1-2 appear to be redundant

Response Point 2:

  • Equations 1-2 are the building blocks used to set where equations 3-7 emanate from, and they are necessary for understanding the model.
  • And equations 3-7 are the specific equations which include the model variables.

  1. Using VAR model is legitimate approach but the notations should be more precise. For instance, as the unit root test rejected the null of stationary, implying that the gross growth rate of GDP per capita is more reasonable to be used by In(GDPK) and growth size of the stock market as In(SMK), among others. These notation changes should be applied to Table 2 and the following tables

Response Point 4:

  • Good observation, but the notations were considered GDPK_1 for the first difference
  • The point noted and the notation change to In(GDPK) in Table 2

  1. Even Table 4 is relevant, the author should be aware of non-linear Grander Causality

Response Point 6:

  • The non-linear granger causality was taken note of through the error term in the model. That is all not captured variables are represesented in the variable.

  1. Liquidity definitions are too sketchy. Chiang and Zheng (2015) provide different definitions of liquidity

Response Point 7:

  • Chiang and Zheng (2015) liquidity definitions were cited and boost the literature in the discussion section and literature review section

Reviewer 2 Report

Dear Author(s),

Please find below my concerns and recommendations regarding your manuscript entiled "Evidence from a Volatile Market on the Impact of the Stock Market on Liquidity and Economic Growth" sent to Economies MDPI Journal.

First of all, during my initial documentation for this review, I found that your manuscript is very similar (more than 85%) to a file sent to Midlands State University. Please clarify this issue and provide some detailed explanations in your answer to this review report.

The article has some issues regarding the administrative organisation: for example, you start with "1. Introduction", and then, the next section is "1.2. Background". But where is "1.1."?!? Please revise and correct this issue.

The Introduction must be improved because it should contain the following important elements:

- the research gap;

- the research goal(s);

- the research question(s).

Please clearly define and describe these elements, because they are very important for the readers.

Within the chapter "2. Economic Growth Theory" you have only a sub-chapter: "2.1. Stock Markets Functions". Please avoid having only one sub-chapter within a main chapter.

Please define a distinct "Literature Review" chapter in your manuscript.

I recommend you to improve the general context of your research by including in your paper the following useful resources: https://doi.org/10.3390/e22050522, http://www.ecoforumjournal.ro/index.php/eco/article/view/884, https://doi.org/10.3390/risks11030060, https://doi.org/10.1016/j.frl.2020.101691. By including these recommended references, you will widen the context of your research proposal.

After the Introduction and Literature Review, you should define and describe the research hypotheses. Every modern scientific article has at least one research hypothesis to be tested.

Before the "6. Concluding Remarks", I recommend you to include a "Discussions" chapter where you present your research results by comparing them to the other results from the scientific literature.

Dear Author(s),

Please consider all the above remarks as being constructive recommendations in order to improve the general quality of your manuscript proposal.

Kind Regards!

Author Response

Response to Reviewer  Comments

Please find below my concerns and recommendations regarding your manuscript entiled "Evidence from a Volatile Market on the Impact of the Stock Market on Liquidity and Economic Growth" sent to Economies MDPI Journal.

Point 1: First of all, during my initial documentation for this review, I found that your manuscript is very similar (more than 85%) to a file sent to Midlands State University. Please clarify this issue and provide some detailed explanations in your answer to this review report.

Response 1:

  • The issue is during the manuscript preparations; we used the Midlands State University Library Turnintin,
  • We submitted the manuscript three times to check for plagiarism; thus, the problem emanated.

Point 2: The article has some issues regarding the administrative organisation: for example, you start with "1. Introduction", and then, the next section is "1.2. Background". But where is "1.1."?!? Please revise and correct this issue.

Response 2:

  • The background became section 1.1, and 1.2 was removed ( 1.1. Background)

Point 3:  The Introduction must be improved because it should contain the following important elements:

- the research gap;

- the research goal(s);

- the research question(s).

Please clearly define and describe these elements, because they are very important for the readers.

Response 3:

  • The research gap- Used an unstable economy characterised by high economic volatility, politically unstable, and a high inflation rate
  • The research goal- as stated in the introduction is to find the link between the stock market and economic growth and liquidity in an unstable economy like Zimbabwe, while there is a positive link in developed countries.
  • The research questions – what is the impact of the stock market in an unstable economies

Point 4: Within the chapter "2. Economic Growth Theory" you have only a sub-chapter: "2.1. Stock Markets Functions". Please avoid having only one sub-chapter within a main chapter.

Response 4:

  • Chapter 2 breaks into a Literature review with a Theoretical review first section and an empirical review second section made as follows
  • Literature review: Theoretical Review
  • 1 Economic growth
  • 2 Stock Market Functions
  • 3 Empirical Review

Point 5: Please define a distinct "Literature Review" chapter in your manuscript.

I recommend you to improve the general context of your research by including in your paper the following useful resources:

 https://doi.org/10.3390/e22050522,

http://www.ecoforumjournal.ro/index.php/eco/article/view/884,

https://doi.org/10.3390/risks11030060,

https://doi.org/10.1016/j.frl.2020.101691.

By including these recommended references, you will widen the context of your research proposal.

Response 5:  

  • The guided references were included in the literature review section, which broadened the subject by, adding literature on stock market volatility issues, and uncertainty effects on the stock market, thus as listed below
  1. Stock Market Volatility and Return Analysis: A Systematic Literature Review
  2. A Profitability Regression Model In Financial Communication Of Romanian Stock Exchange's Companies
  3. Paper Title - A Conceptual Model of Investment-Risk Prediction in the Stock Market Using Extreme Value Theory with Machine Learning: A Semi Systematic Literature Review
  4. The Impact of COVID-19 on emerging stock markets

Point 6: After the Introduction and Literature Review, you should define and describe the research hypotheses. Every modern scientific article has at least one research hypothesis to be tested.

Response 6:

  • The Research Hypothesis

H0: A positive relationship exists between the stock market and economic growth in unstable economies.

H0: A positive relationship exists between the stock market and liquidity in unstable economies.

H1: A negative relationship exists between the stock market and economic growth in unstable economies.

H1: A negative relationship exists between the stock market and liquidity in unstable economies.

Point 7: Before the "6. Concluding Remarks", I recommend you to include a "Discussions" chapter where you present your research results by comparing them to the other results from the scientific literature.

Response 7: 

  • Discussion chapter added, chapter 6

Reviewer 3 Report

 It is an interesting paper to determine if the impacts of the stock market are communicated in an economically unstable environment characterized by volatility, high inflation rates, and political instability. Therefore, I have read the complete paper and hold the following observation.

  1. The paper is well written. However, further proofreading will improve the manuscript.
  2. More discussion on the research contribution and knowledge gap needs to be added with the introduction and linked to the research questions and motivation for sample selection. This is also important in giving the study's motivation to the reader. 
  3. The literature needs to be expanded; for instance, this study (Regression Analysis of Macroeconomic Conditions and Capital Structures of Publicly Listed British Firms. Mathematics 2022, 10, 1119 ) could be analyzed and added to expand the literature review and the results sections as they have similar variables.
  4. More discussion of descriptive statistics should be added. The paper could gain a lot in terms of readability if further sample discussion could be added to the discussion and conclusion sections. 
  5. The equations need to be revised and explained further. For instance, the reason for making the variables in bold font and some letters are not in bold font.
  6. Table 1 need to be revised (some reported observation is in bold, any reason for that?).
  7. The methodology and results could have been further discussed and linked to the previous research that has used similar methods or considered other factors such as structural breaks and financial crises. 
  8. It would be essential to see some robustness/sensitivity checks beyond what the authors have done. Have any robustness checks been done to confirm the validity of their findings?
  9. The conclusion needs further improvement and more discussion on the research limitation and policy implications.

I wish you all the best.

Author Response

Response to Reviewer 3 Comments

It is an interesting paper to determine if the impacts of the stock market are communicated in an economically unstable environment characterised by volatility, high inflation rates, and political instability. Therefore, I have read the complete paper and hold the following observation.

Point 1: The paper is well written. However, further proofreading will improve the manuscript.

Response 1:

  • Noted, and an English native reader proofread the manuscript and also with the use of Grammarly software editor

Point 2: More discussion on the research contribution and knowledge gap needs to be added with the introduction and linked to the research questions and motivation for sample selection. This is also important in giving the study's motivation to the reader. 

Response 2:

  • The introduction of added the knowledge gap and the research questions

Point 3: The literature needs to be expanded; for instance, this study (Regression Analysis of Macroeconomic Conditions and Capital Structures of Publicly Listed British Firms. Mathematics 2022, 10, 1119 ) could be analysed and added to expand the literature review and the results sections as they have similar variables.

Response 3:

  • More literature review was added to the manuscript in the literature review section and background

Point 4: The equations need to be revised and explained further. For instance, the reason for making the variables in bold font and some letters are not in bold font.

Response 4:

  • the issue with the bold font was typo errors; there were no reasons for differences in fonts. Therefore, the problem was resolved by making the variables in bold font to be not in bold font.
  • The variables are in bold font for distinction from other words in the article, trying to show that these are the variables used in the study, for example –economic growth as the primary variable under study.

Point 5: Table 1 need to be revised (some reported observation is in bold, any reason for that?).

Response 6:

  • In table 1, the bold removed

Point 6: The methodology and results could have been further discussed and linked to the previous research that has used similar methods or considered other factors such as structural breaks and financial crises. 

Response 6:  

  • A chapter for discussion was introduced, which linked the current and previous research similarities

Point 7: It would be essential to see some robustness/sensitivity checks beyond what the authors have done. Have any robustness checks been done to confirm the validity of their findings?

Response 8:

  • The robustness checks were done using diagonistics tests. After the estimation and testing of the time series model with VAR, the results were diagnostically tested for the model's autocorrelation, stability, and reliability to check the suitability and reliability of the model and results.

Point 9: The conclusion needs further improvement and more discussion on the research limitation and policy implications.

Response 9:

  • The conclusion was improved by adding more discussion on the research limitations and policy implications

Round 2

Reviewer 2 Report

Dear Author(s),

After reading the new version of your manuscript proposal, I have the following recommendations and concerns:

1. At the rows 344 -352, the font seems to be different from the rest of the text. Please revise and correct it.

2. At the rows 344 - 352, there are two distinct hypotheses with the same number: H0 appears twice, H1 appears twice. Please revise and correct this issue.

3. The background and the literature review chapters still should be improved by including the following additional resources: https://doi.org/10.3390/su9030335, http://www.ecoforumjournal.ro/index.php/eco/article/view/884, https://doi.org/10.3390/jrfm12040176, https://doi.org/10.3390/jrfm13090208.

4. In the Discussion chapter, please state clearly if your research hypotheses are supported or not. The goal of a scientific article is to test research hypotheses.

5. In the "7. Concluding Remarks" chapter, I recommend you to also include the following important aspects:

- the managerial implications (here is the place where you can "sell" your research results to the readers);

- the research limitations;

- the future research directions.

Kind Regards!

Author Response

After reading the new version of your manuscript proposal, I have the following recommendations and concerns:

  1. At the rows 344 -352, the font seems to be different from the rest of the text. Please revise and correct it.

 Response 1

  • The rows 344-352 font corrected
  1. At the rows 344 - 352, there are two distinct hypotheses with the same number: H0 appears twice, H1 appears twice. Please revise and correct this issue.

 Response 2

  • The issue of H0 was revised and corrected
  1. The background and the literature review chapters still should be improved by including the following additional resources:

 https://doi.org/10.3390/su9030335,

http://www.ecoforumjournal.ro/index.php/eco/article/view/884,

 https://doi.org/10.3390/jrfm12040176,

 https://doi.org/10.3390/jrfm13090208

Response 3

  • The background was improved by including the above literature
  • Lines (61-63)- (74-77)- (166-168)

  1. In the Discussion chapter, please state clearly if your research hypotheses are supported or not. The goal of a scientific article is to test research hypotheses.

 Response 4

  • The support of the research hypothesis was added in the discussion section
  1. In the "7. Concluding Remarks" chapter, I recommend you to also include the following important aspects:

- the managerial implications (here is the place where you can "sell" your research results to the readers);

- the research limitations;

- the future research directions.

Response 5

  • The managerial implications are captured on the contribution of the study whereby we advise the importance of the research to the policymakers
  • The research limitations – are added last section -lines 685-689
  • Further research directions – also added on the last section lines 690-692

Kind Regards!

Round 3

Reviewer 2 Report

Dear Author(s),

Please see below my remarks for your revised version of the article.

The H0 Hypothesis appears several times in your manuscript. The first apparition of "H0" is at the row 354 (research hypothesis). Then, you also have "H0"s at the rows 473, 558, 561, 563. These "H0"s can be confusing for the reader because the reader expects to follow the evolution of the research hypothesis. Please find a manner to avoid this potential confusion.

I recommend you to include additional relevant references from the scientific literature in your work.

Starting from the current research results and the research limitations, the future research directions must be developed in the Conclusions chapter.

In the Conclusions chapter, please also include the managerial implications.

Kind Regards!

Author Response

  1. The H0 Hypothesis appears several times in your manuscript. The first apparition of "H0" is at the row 354 (research hypothesis). Then, you also have "H0"s at the rows 473, 558, 561, 563. These "H0"s can be confusing for the reader because the reader expects to follow the evolution of the research hypothesis. Please find a manner to avoid this potential confusion.

Response 1

  • Alternative writing of H01 was introduced that is line 354 ,
  • Row 473 adapted H02
  • Row 558, 561, 563 USED H03

  1. I recommend you to include additional relevant references from the scientific literature in your work.

Response 2.

  • Rows 314 -326 new literature added
  • Discussion chapter also

  1. Starting from the current research results and the research limitations, the future research directions must be developed in the Conclusions chapter.

Response 3

  • Conclusions chapter, started with current research results, research limitatiosn and future research directions; the corrections made in the chapter

  1. In the Conclusions chapter, please also include the managerial implications.

response 4

  • Lines 680-688 added crucial managerial implications

Kind Regards!